# Multi-Prismatic Hollow Cube CeVO_4_ with Adjustable Wall Thickness Directed towards Photocatalytic CO_2_ Reduction to CO

**DOI:** 10.3390/nano13020283

**Published:** 2023-01-10

**Authors:** Yong Zhou, Guan Wang, Jiahao Wu, Zihao Chen, Chen Zhang, Ping Li, Yong Zhou, Wei Huang

**Affiliations:** 1School of Flexible Electronics (SoFE) & Institution of Advanced Materials (IAM), Nanjing Tech University, 30 South Puzhu Road, Nanjing 211816, China; 2School of Physical and Mathematical Sciences, Nanjing Tech University, 30 South Puzhu Road, Nanjing 211816, China; 3College of Food Science and Light Industry, Nanjing Tech University, 30 South Puzhu Road, Nanjing 211816, China; 4National Laboratory of Solid State Microstructures, Department of Physics, and Collaborative Innovation Center of Advanced Microstructures, Nanjing University, Nanjing 210093, China

**Keywords:** hollow cubic, template method, vanadate-based photocatalyst, tetragonal phase, CO_2_ reduction

## Abstract

Ternary orthovanadate compounds have received increasing attention due to their broad light absorption and diverse crystal structure. However, their multi-assembled crystal morphologies are limited mainly due to their initially polyatomic VO_4_ groups. In this study, multi-prismatic hollow cubic CeVO_4_ microstructures were fabricated by a one-step solvothermal method without any organic agents. The increase in wall thickness is in accordance with the radial direction of the quadrangular prism. Moreover, the overdose of the V precursor is favorable for the formation of hollow micro-cubic CeVO_4_, and the wall thickness changes from 200 to 700 nm. Furthermore, these CeVO_4_ microstructures were applied to photocatalytic CO_2_ reduction with a maximum CO generation rate of up to 78.12 μmol g^−1^ h^−1^ under visible light irradiation, which was several times higher than that of the other samples. This superior photocatalytic activity might be attributed to its good crystallinity and unique exposed interior structure. This study provides guidelines for the multi-assembled structure fabrication of ternary compounds and expands upon the exploration of the spatial structure of multivariate compounds.

## 1. Introduction

Global CO_2_ emissions have had a significant impact on the ecosystem due to climate change, and atmospheric concentrations have broken the 400 ppm threshold [1]. This continued increase in emissions incentivizes green and recyclable approaches to reduce atmospheric CO_2_ concentrations [2,3]. The photocatalytic CO_2_ reduction, as an environmentally friendly and sustainable route, directly utilizes renewable solar energy to convert CO_2_ molecules into carbon-based compounds and O_2_ without any other byproducts [4]. In this process, photocatalysts play an irreplaceable role in the redox reaction as the main light-absorbing materials.

Ternary orthovanadate compounds (MVO_4_), as one promising material, have been extensively investigated to realize solar energy utilization due to their broad light absorption properties and superior surface atomic arrangement [5]. Among them, the light absorption edge of the tetragonal CeVO_4_ material is extended to about 800 nm [6], which accounts for about half of the radiant energy of sunlight. The broad wavelength distribution of sunlight provides the requirements for the light absorption range of MVO_4_. The “scaffold” of VO_4_ enables the regulation of the hybridization of the M outmost and V 3d orbitals, which plays an important role in determining the band position [7]. However, the dominantly polyatomic VO_4_ groups contain five atoms with a spatially distributed tetrahedral structure, which restricts the construction of multi-assembled crystal structures [8]. In addition, for the CeVO_4_ semiconductor, it usually appears in the tetragonal structure with high crystal symmetry (I41/amd). Furthermore, it contains CeO_8_ groups, which are made of a Ce ion and eight coordinated O ions [9]. The periodic arrangement of the atoms results in various morphologies, with sizes ranging from nanometers to micrometers. The CeVO_4_ nanoparticles were obtained in a liquid solution and exhibited excellent photocatalytic activity under visible light [10]. Moreover, CeVO_4_ nanorods easily form with multi-ion involvement in the precipitation solution [11,12]. These structures were further assembled as hollow microspheres [13,14]. Generally, the morphologies of CeVO_4_ tend to comprise one-dimensional structures with a smooth surface, and artificial microstructures with abundant edges require urgent exploration.

In addition, among various methods, template synthesis, as a relatively simple and easy route for fabricating assembled structures, has garnered strong interest due to its high thermal stability and good narrow interdomain limiting effects [15]. The templates greatly impact the size and morphology of the prepared products. The NaCl, as an inexpensive and environmentally friendly material, was used as a template in a non-aqueous system [16]. The precursor molecules are easily wrapped on the surface of NaCl because the energy barrier relative to nucleation can be substantially reduced [17], and the template can also be easily removed by using washing processes with deionized water. Bao et al. successfully prepared Ni single-atom catalysts bonded to the oxygen sites on graphene-like carbon (Ni-O-G SACs) with NaCl as the template, and Ni-O-G SACs exhibited a 3D porous framework with a simultaneous NaCl-like hollow cube shape [18]. Additionally, Zhang et al. demonstrated that the edges, corners, and surfaces of NaCl are favorable for the nucleation and growth of metallic transition metals, with the dichalcogenides nanosheet as the template [19]. Further, NaCl was certified as being able to increase the porosity of graphitic carbon nitride pyrolysis and to effectively improve the oxygen reduction reaction activity of the catalyst as a hard template [20]. Most reports focused on metal- or carbon-based compounds using the template method, probably due to their smaller nucleation cells [21,22]. The application of this method to ternary compounds with broad light absorption is urgently required, as it is beneficial for solar energy utilization and further photocatalytic efficiency improvements.

In this study, the novel hollow cubic CeVO_4_ microstructures were fabricated by the solvothermal method with NaCl as the template, which provided the preferential sites for nuclei and crystal growth and also stabilized the cubic structure. In addition, each cube contains numerous quadrangular prisms as the secondary structure, stacked along the longitudinal edges. The thickness of the cubic wall increases as the precursor of the vanadium source increases from 200 to 700 nm with the longitudinal growth of the prisms. These prepared CeVO_4_ hollow cubes exhibited photocatalytic CO_2_ reduction performance, and the maximum CO generation rate was 78.12 μmol g^−1^ h^−1^ for CV2 under visible light irradiation. The precursors are within the stoichiometric ratio for CV2, contributing excellent crystallinity and continuous crystal growth. The CV1 and CV3 exhibited CO evolution rates of 14.28 and 11.22 μmol g^−1^ h^−1^, respectively. Their lower efficiency compared to CV2 might be attributed to the uniform structure resulting from the unbalanced proportion of precursor solutions.

## 2. Experimental Section

### 2.1. Materials

All precursors used in this study were of analytical grade without any purification processes. The manufacturers and purity of the precursors were as follows: CeCl_3_•7H_2_O (Macklin, 99.9%, Shanghai, China), NH_4_VO_3_ (Sinopharm, 99.0%, Beijing, China), CH_3_CH_2_OH (Wuxiyasheng, 99.7%, Wuxi, China), and NaCl (Aladdin, 99.5%, Shanghai, China).

### 2.2. Synthesis of Hollow Cubic CeVO_4_ Microstructure

The synthetic process of CeVO_4_ is displayed in Figure 1. Initially, 4 mL of saturated NaCl was added dropwise to 10 mL CH_3_CH_2_OH during the mixing process. Subsequently, the precursors of 1 mmol CeCl_3_•7H_2_O and 1 mmol NH_4_VO_3_ were added into the prepared solution. The prepared solution was stirred for 15 min at room temperature and then transferred to an autoclave heating at a temperature of 160 °C for 24 h. After the reaction, the obtained samples were washed three times with ultrapure water and ethanol. Finally, a brown powder was obtained by centrifugation, filtration, and freeze-drying processes and denoted CV2. In addition, the experiment used different molar ratios (1:0.75 and 1:1.25) with respect to the precursors, referred to as CV1 and CV3, respectively.

### 2.3. Characterization

The nano- and micro-structural morphologies and elemental information on the samples were characterized by scanning electron microscopy (SEM; JSM-7800F, JEOL, Tokyo, Japan) and high-resolution transmission electron microscopy (HRTEM; JEM 200CX, JEOL, Tokyo, Japan). Energy-dispersive spectroscopy (EDS) was used to determine the composition types and contents of the samples. The X-ray diffraction (XRD) patterns were obtained using a SmartLab (3 KW) instrument operating with a Cu Ka source within a 2θ range from 10° to 90° at a scan rate of 20° min^−1^ to identify the phase structure of the samples (Rigaku Ultima III, Tokyo, Japan). UV–visible diffuse reflectance spectra (UV–Vis DRS) were obtained at room temperature using an ultraviolet–visible spectrophotometer and directly converted to absorption spectra by using the Kubelka–Munk equation in the software (UV-2550, Shimadzu, Kyoto, Japan). The time-resolution photoluminescence (TRPL) spectra were used to investigate the time-dependent kinetics of the excited-state radiation leap spectra of cerium vanadate (FSL980-STM, Edinburgh, UK). The information on the vibrational and rotational aspects of CeVO_4_ was derived via Raman spectroscopy using the Raman scattering effect, and we investigated its molecular structure (iHR550, Horiba, France). Our group used the PSTrace5 portable workstation to obtain access to electrochemical impedance spectroscopy (EIS). The Tafel data were obtained by using another electrochemical workstation (CHI66E, Shanghai, China). All electrochemical characterizations were accomplished on a standard three-electrode system. The electrolyte was a 0.5 mol L^−1^ Na_2_SO_4_ solution. The working electrode was prepared by depositing the catalyst on F-doped SnO_2_ conductive glass (FTO). The reference electrode was Ag/AgCl, and Pt was used as the counter electrode. The specific surface area was calculated based on the Brunauer–Emmett–Teller (BET) adsorption model (ASAP 2460, Micromeritics, Georgia, GA, USA).

### 2.4. Photocatalytic CO_2_ Reduction

The photocatalytic reaction of carbon dioxide reduction was carried out at room temperature and pressure, and a 0.01 g sample of the prepared powder was evenly weighed on top of a glass reactor with an area of 4.91 cm^2^ and a volume of 420 mL. The bottom of the glass reactor was tied to a magnetic stirrer. Moreover, the reaction of pure CO_2_ gas (99.99% purity) was guaranteed by slow and even injections until it filled the reactor. The excess gas was collected with the NaHCO_3_ solution, and finally, the glass reactor was placed in a gas-tight reaction system with a volume of about 440 mL. In total, 0.4 mL of pure water was injected from the silicone rubber diaphragm into the reactor via a micro-syringe as the reducing agent and further injected into the reaction system. After stirring for 1 h, the adsorption of the CO_2_–H_2_O atmosphere was balanced to ensure complete adsorption. A solar simulator (Microsolar 300W xenon lamp) was used to illuminate the vertical sample. Consequently, from the start of the xenon lamp, the lighting time was 5 h. Around 1 mL of gas was drawn from the reaction cell. After injections, a gas chromatograph (GC-9860-5C-NJ, Hao Erpu, China) was used to analyze the subsequent reaction products. The column, carrier gas, and detector used for GC comprised porapak Q and N, Ar gas, and FID, respectively.

## 3. Result and Discussion

### 3.1. Catalyst Characterization

The crystal phase of the as-prepared samples was analyzed by X-ray powder diffraction (XRD). Appendix A shows the XRD patterns of the nanocomposites and the CeVO_4_ reference. All the diffraction peaks of the products obtained from different molar ratios of the precursors were assigned to the tetragonal phase of CeVO_4_ (JCPDS No. 12-0757). The XRD patterns of CeVO_4_ showed four main diffraction peaks at 2θ = 18.126, 24.032, 32.399, and 47.860 corresponding to the (101), (200), (112), and (312) planes, which also indicated a tetragonal crystal system of CeVO_4_ with space group I41/amd (141) and cell parameters a = 7.339 Å, b = 7.339 Å, c = 6.496 Å, α =90.0°, β = 90.0°, and γ = 90.0°. From the XRD patterns, no other characteristic impurities were found, indicating that the material contained only pure CeVO_4_. 

The hollow cubic CeVO_4_ microstructures were prepared using sodium chloride as a template by using the one-step solvothermal method. In particular, CV1, CV2, and CV3 were obtained by adding the precursors cerium chloride heptahydrate and ammonium metavanadate at molar ratios of 1:0.75, 1:1, and 1:1.25, respectively. Scanning electron microscopy (SEM) images show that CeVO_4_ hollow cubes monodispersed in isolation with a size of approximately 2 µm for CV2, and the proportion of complete cubes was over 90% (Figure 2a–c and Appendix A). There were clear edges for each cube, which were assembled with quadrangular prisms. These prisms mostly “stood” in rows without any gaps on the facets of the cube, and their orientation changed with the curvature of the edges of the cube. The precursors dissolved in the NaCl solution, and vanadate ions were spontaneously immobilized in atomic clusters in solution during the crystallization process shown in Figure 2d. During the natural cooling process, crystal grains formed preferentially on the edges of the NaCl cube due to the lower energy barrier, and then these crystals continued to grow. The elements Ce, V, and O were uniformly distributed on the hollow cubes when the grains grew (Figure 2e–g). However, a lower number of nucleation sites were thermodynamically distributed on the centers of the facets of the cube, which might have resulted in the formation of hollow structures. In the nucleation process for CeVO_4_, this can be traced back to the smaller structural units, namely, the VO_4_ and CeO_8_ groups [23]. In order to explore their effects, the molar ratios of the precursors were modulated. For CV1, nanoparticles assembled in the walls with curvature, and the thickness was about 200 nm, which is comparable to the size of nanoparticles (Appendix A). The limited nanoparticle growth might be attributed to the lower concentration of the V precursor compared to that in CV2 (wall thickness of about 500 nm), which possessed abundant VO_4_ groups for nucleation and crystal growth. The excessive VO_4_ groups made the cubic wall thicker, about 700 nm, and the substructure of the columnar body grew taller for CV3 (Appendix A). In addition, a few isolated prisms were generated. In order to detect the growth direction of the quadrangular prims as the substructure of the hollow cube, high-resolution transmission electron microscopy (HRTEM) characterization was implemented (Appendix A). The two main lattice spacings at 0.48 and 0.37 nm were observed for CV2, which are attributed to crystal planes (101) and (200) for the tetragonal CeVO_4_, respectively. The angle of the two crystal planes is about 48.8°, which is consistent with the theoretical value of 48.7°, and it is the same as the angle between their crystal orientations ([101] and [200]). Notably, the growth orientation of quadrangular prims is perpendicular relative to [200] and at a 41.2° angle relative to [101]. As a result, the prism grows along [001] and continues to grow with the abundant supply of VO_4_ groups.

The formation process of the hollow cube was investigated by varying reaction times at intervals of four hours for CV2. The first stage for the formation of hollow cubes is nucleation, and nuclei are inclined toward forming on the NaCl cube discretely due to the lower energy barrier relative to nucleation [24]. The distribution of nuclei is greatly separated due to their limited number, and simultaneously, these nuclei cannot grow to be sufficiently large after a 4 h reaction (Figure 3a). Thus, the nanoprisms scattered when the NaCl cube was removed. The number of nuclei increased with the reaction time up to 8 h, and sub-nanoprisms partially assembled along the edges of NaCl (Figure 3b). The continuous stacking of nanoprisms contributes to the formation of hollow cubes and additionally promotes crystal growth with [001] orientation due to anisotropic interfacial tension [25]. The framework of the hollow cube formed after a 12 h reaction, but it was fragile and collapsed when NaCl was removed (Figure 3c). Then, the strength of the hollow cube increased as the reaction time increased to 16 h (Figure 3d) due to the growth of nanoprisms. The entire hollow cube retained its full shape, and the substructure of nanoprisms grew along the [001] orientation, probably due to the three-way interface contact restriction (Figure 3e) [26]. The width of the edges increased slightly for CV2 as the reaction time increased to 24 h, and it retained the complete hollow cube (Figure 3f). The flow chart of the growth processes is illustrated in Figure 3g. The formation of the hollow cube is closely dependent on the nucleation and crystal growth processes. 

The information about the vibration and rotation transitions of CeVO_4_ molecules was further provided by Raman spectroscopy (Figure 4a). Five peaks of CeVO_4_ were detected under the excitation of a laser with an excitation wavelength of 532 nm. The Raman peaks at 245 cm^−1^ were assigned to the external mode of Ce–VO_4_ vibration [27], which could prove the formation of CeVO_4_ materials. The A_1g_ and B_1g_ bending modes of VO_4_^3−^ were observed at 356 cm^−1^ and 448 cm^−1^, respectively. The anti-symmetrical stretching (B_1g_) of VO_4_^3−^ presented at the 756 cm^−1^ Raman peak, and symmetric stretching (A_1g_) appeared at the 827 cm^−1^ characteristic peak [28]. The nitrogen adsorption–desorption isotherm analysis was applied to determine the specific surface area of different samples by Brunauer–Emmett–Teller (BET) characterization (Figure 4b). The three isothermal curves displayed the same type of IV feature [29]. Further, CV2 showed a specific surface area of 14.43 m^2^ g^−1^, which is a little larger than those of CV1 and CV3 at 13.28 m^2^ g^−1^ and 12.68 m^2^ g^−1^, respectively.

The UV–Visible diffuse reflectance spectra (UV–Vis DRS) were employed to further investigate the optical absorption properties and bandgap of the prepared CeVO_4_ samples. As shown in Figure 4c, the absorption edges of all series of pure CeVO_4_ were located at about 800 nm with extended visible light absorptions. However, the light absorption range of CeVO_4_ redshifted a little at first and then blueshifted as the Ce/V ratio increased. The light absorption ability of CV2 was slightly stronger than that of CV1 and CV3. Furthermore, the corresponding bandgaps of the samples were calculated by the transformed Kubelka–Munk function. The band gaps were determined as 1.62 eV, 1.60 eV, and 1.66 eV for CV1, CV2, and CV3, respectively (Appendix A). In addition, CeVO_4_, as an n-type semiconductor, shows redox capability due to its suitable conduction and valence band positions [30,31,32].

Time-resolved fluorescence decay spectroscopy measurements were performed at an excitation wavelength of 420 nm to investigate the photoexcited charge carrier transfer behavior of CeVO_4_. Figure 4d showed that the lifetime of CV2 is 2.30 ns, which is substantially longer than those of CV1 (1.78 ns) and CV3 (1.70 ns). However, for an n-type semiconductor with a direct leap, the main factor determining the carrier’s lifetime is the direct recombination process of the conduction band’s electrons and valence band’s holes, due to the fact that both the bottom of the conduction band and the top of the valence band are at the same point in the Brillouin region. Therefore, the carrier lifetime of CeVO_4_ is generally relatively short. We also used electrochemical impedance spectroscopy (EIS) to further analyze the effect of the Ce/V ratio on the charge migration kinetics. The results of the EIS-Nyquist plot are shown in Figure 4e. Apparently, the lower impedance of the CV2 material compared to those of CV1 and CV3 implies a faster transfer of interfacial charges within CV2 materials, resulting in an effective separation of the e^−^/h^+^ pairs. Moreover, the Tafel data in Figure 4f also demonstrate that CV2 exhibits kinetic advantages in the catalytic process, with smaller Tafel slope values of 216.82 dec^−1^ compared to 265.49 and 236.05 mV dec^−1^ for CV1 and CV3, respectively.

### 3.2. Photocatalytic Performance for CO_2_ Reduction

The photocatalytic CO_2_ reduction performance was evaluated in a gaseous reactor under visible light irradiation for the hollow cubic CV samples. CO, as the main product, was detected, and its yield was recorded every hour within a reaction period of 5 h. As shown in the equation CO_2_ + 2H^+^ + 2e^−^ → CO + H_2_O (E^0^ = −0.53 eV vs. NHE pH = 7), the formation of the CO product requires only two electrons and two protons with a reduction potential of −0.53 V vs. NHE, which is supposed to be easily attainable as the main product [33]. The maximum CO evolution amount was 390.59 μmol g^−1^ for CV2, compared to 71.38 μmol g^−1^ and 56.10 μmol g^−1^ for CV1 and CV3, respectively (Figure 5a). The corresponding CO evolution rate was 78.12 μmol g^−1^ h^−1^ for CV2—about 5.5 and 7 times the values for CV1 and CV3, respectively (Figure 5b). In contrast, no CO was detected without a photocatalyst or light.

The hollow cube structure can successfully drive CO_2_ reduction to CO without any other gaseous products under visible light irradiation (Figure 5c). Including favorable reaction kinetic conditions [34], the porous microstructure improves the contact between CO_2_ and the catalyst. The precursors of V and Ce with stoichiometric ratios are advantageous for photocatalysis, which might be attributed to their excellent structure with high symmetry and the fine crystal growth environment. While excessive VO_4_ groups contribute to the formation of hollow cubes, photocatalytic efficiencies decrease. The boundary effect of nanoprisms that still have space to grow can probably explain these results. This study provides guidelines for preparing photocatalysts, especially with respect to hierarchical structure.

## 4. Conclusions

In summary, novel hollow cubic CeVO_4_ nanomaterials were fabricated by a one-step solvothermal method for enhanced photocatalytic CO_2_ reduction to CO. The excessive addition of V precursors contributed to the formation of hollow cubes by improving the growth of the substructure. The quadrangular prisms that grew along the [001] direction were assembled into cubes. In addition, VO_4_, as one constituent unit of CeVO_4_, contributed to the growth of prims. Moreover, optimal CO was generated during photocatalytic CO_2_ reduction with CV2 at a precursor ratio of 1:1 for Ce and V. The maximum CO yield rate was 78.12 μmol g^−1^ h^−1^, which is 5.5 and 7 times higher than those for CV1 and CV3, respectively. This improved efficiency might be attributed to the porous structure and excellent crystallinity. This study reveals the prospect of modulating photocatalyst arrangement by using the template method, with potential applications for other ternary compounds, which should greatly improve photocatalytic efficiency and product selectivity.

## Figures and Tables

**Figure 1 nanomaterials-13-00283-f001:**
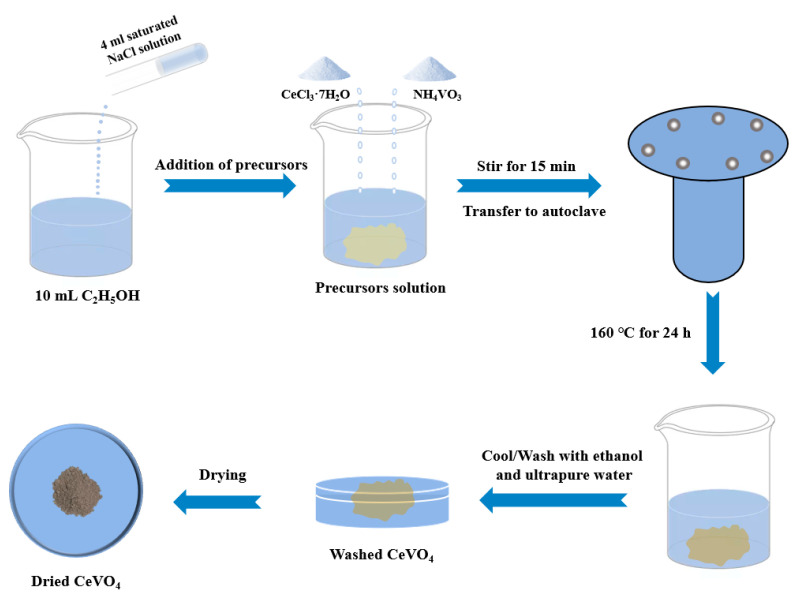
Schematic diagram of the synthetic process of hollow cubic CeVO_4_.

**Figure 2 nanomaterials-13-00283-f002:**
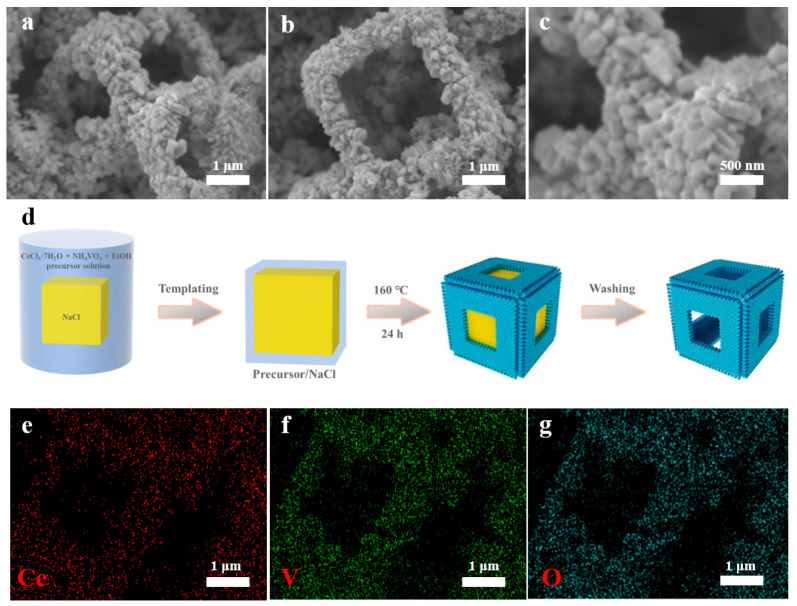
SEM images of CV2 from different angles (**a**,**b**); enlarged SEM image of (**c**); schematic illustration of the synthesis of CV2 (**d**); elemental mapping images of Ce (**e**), V (**f**), and O (**g**) of CV2.

**Figure 3 nanomaterials-13-00283-f003:**
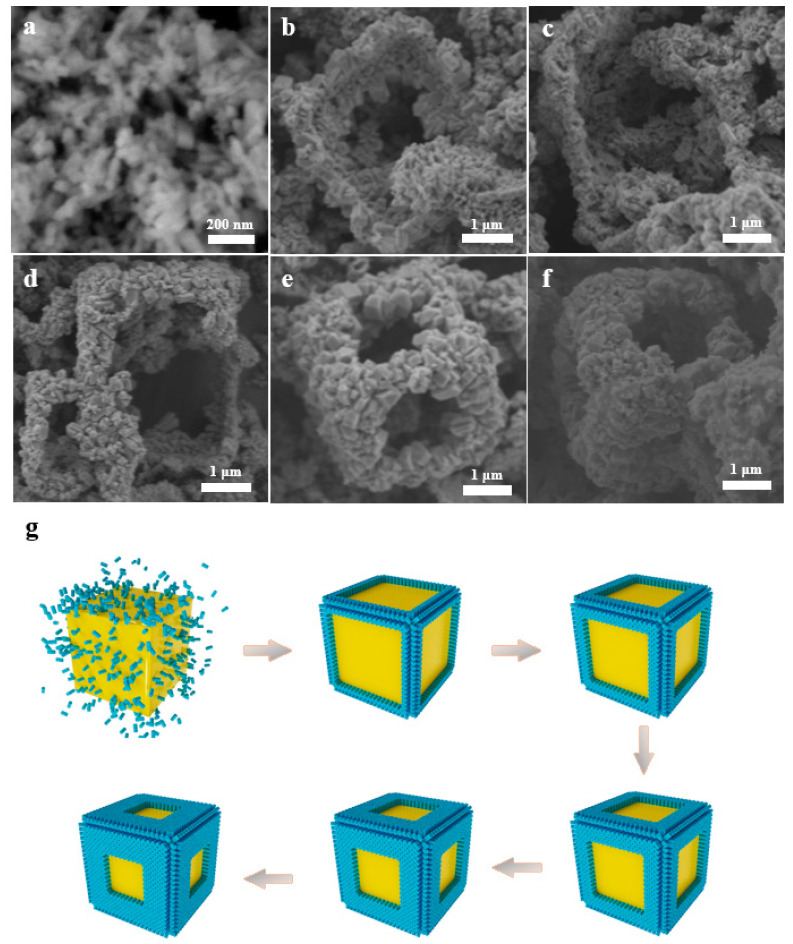
SEM images for CV2 prepared by solvothermal reactions at 4 h (**a**), 8 h (**b**), 12 h (**c**), 16 h (**d**), 20 h (**e**), and 24 h (**f**), and the flow chart of growth processes for the CeVO_4_ hollow cube (**g**).

**Figure 4 nanomaterials-13-00283-f004:**
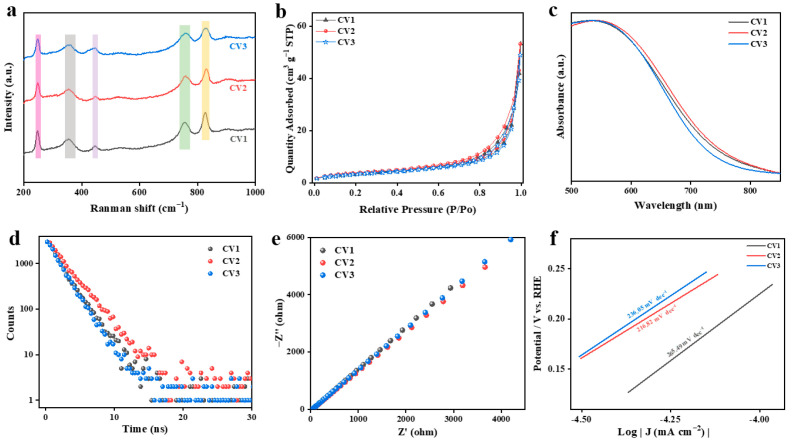
(**a**) Raman scattering spectroscopy. (**b**) Nitrogen adsorption-desorption isotherms. (**c**) UV-Vis DRS. (**d**) Photoluminescence spectra. (**e**) Electrochemical impedance spectroscopy (EIS) Nyquist plots. (**f**) Tafel curves for CV1, CV2, and CV3.

**Figure 5 nanomaterials-13-00283-f005:**
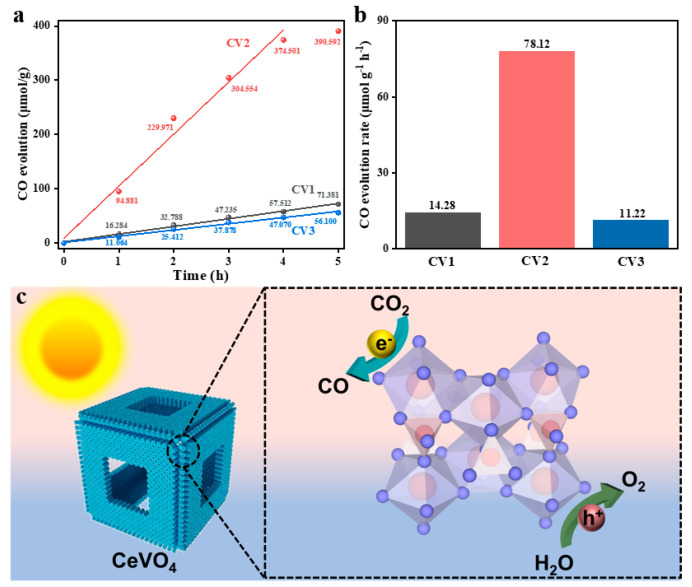
(**a**) Photocatalytic CO production amount and (**b**) corresponding CO evolution rate for CV1, CV2, and CV3. (**c**) Schematic illustration of photocatalytic mechanism for CeVO_4_ under sunlight irradiation.

## Data Availability

Not applicable.

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
