# Peer review of "Multi-Prismatic Hollow Cube CeVO4 with Adjustable Wall Thickness Directed towards Photocatalytic CO2 Reduction to CO"

_nanomaterials, 2023, doi:10.3390/nano13020283_

Round 1

Reviewer 1 Report

The article demonstrates the original technique for the formation of CeVO4 semiconductor photocatalyst with a highly developed surface and the results of its application for CO2 reduction. NaCl cubic crystals are used as a template. The authors were able to achieve that CeVO4 was formed on the edges of the NaCl cubes. Therefore, after dissolving NaCl crystals in water, the authors obtained CeVO4 in the form of hollow cubes. As a result, the material acquired a highly developed surface. Thus, the authors have formed CeVO4 semiconductor material that is photoactive in the visible region of the spectrum, which is favorable for gaseous photocatalysis using sunlight. The authors demonstrated the efficiency of the obtained semiconductor material for the photocatalytic reduction of CO2. The results are interesting. The article may be accepted for publication.

Author Response

Reviewer #1 Review report "Multi-prismatic Hollow Cube CeVO4 with Adjustable Thickness Towards Photocatalytic CO2 Reduction to CO"

Comments: The article demonstrates the original technique for the formation of CeVO4 semiconductor photocatalyst with a highly developed surface and the results of its application for CO2 reduction. NaCl cubic crystals are used as a template. The authors were able to achieve that CeVO4 was formed on the edges of the NaCl cubes. Therefore, after dissolving NaCl crystals in water, the authors obtained CeVO4 in the form of hollow cubes. As a result, the material acquired a highly developed surface. Thus, the authors have formed CeVO4 semiconductor material that is photoactive in the visible region of the spectrum, which is favorable for gaseous photocatalysis using sunlight. The authors demonstrated the efficiency of the obtained semiconductor material for the photocatalytic reduction of CO2. The results are interesting. The article may be accepted for publication.

Reponse: We thank the Reviewer for his/her recommendation, which are very helpful for us to further improve our manuscript.

According to your advice, this manuscript was edited for proper English language, grammar, punctuation, spelling, and overall style by native English-speaking editors on MDPI.

Reviewer 2 Report

The manuscript deals to prepare and investigate hollow cubic CeVO4 microstructures. These structures were applied to photocatalytic reduction of CO2.  The subject is interesting; it is related to the profile of the journal. The experimental design is appropriate, several up-to-date methods and equipment were used to prove the results. The manuscript is well written, clear, and understandable, figures are nice and supporting. The English of the manuscript seems to be good and understandable. (The referee is not a native English speaker).

Comments:

  1. The Title is informative.
  2. The Abstract reflects the approach of the study, nevertheless, the main results of the work also should be presented here.
    • The unit umol/g cat h is a little bit confusing. It is clear, that “cat” means the catalysts, but it is seems as a unit. g(cat) or other may be more understandable.
  3. The section Introduction presents the important points of the topic, contains references related to the work, and reveal the importance and novelty of the work.
    • p 2, end of paragraph 2: “… while plastic microstructures …” Why do you mention plastics?
    • The mechanism of CO2 reduction should be presented in more detailed.
  4. The section Experimental.  The experimental design is appropriate; it is generally well described. However, the experimental circumstances should be presented in more detailed in all cases; moreover, the description of photocatalytic reactor also should be improved. What was the volume of the reactor, what was the pressure? What was the column, carrier gas and the detector of the GC? What was the composition of the CO2 containing gas? Was it pure CO2?
  5. In the section Results and discussion, authors describe the results shown in the corresponding figures, and tables. The figures are nice and informative.
    • Have you detected other products than CO?
    • Could you provide a more detailed reaction mechanism or equations? What is the fate of hydrogen (from water)?
  6. The Conclusions section well summarizes the result of the work, however, practical applicability or importance also should be presented.
  7. Conclusion and recommendation: This manuscript is recommended for publication after minor revision.

Author Response

Reviewer #2 Review report " Multi-prismatic Hollow Cube CeVO4 with Adjustable Thickness Towards Photocatalytic CO2 Reduction to CO"

Comments: The manuscript deals to prepare and investigate hollow cubic CeVO4 microstructures. These structures were applied to photocatalytic reduction of CO2.  The subject is interesting; it is related to the profile of the journal. The experimental design is appropriate, several up-to-date methods and equipment were used to prove the results. The manuscript is well written, clear, and understandable, figures are nice and supporting. The English of the manuscript seems to be good and understandable. (The referee is not a native English speaker).

Reponse: We thank the Reviewer for the following comments, which are very helpful for us to further improve our manuscript. We revise the whole manuscript and make changes for these specific points.

All changes are marked with “Track Changes” in the manuscript.

  1. The Title is informative.

Reponse: Once again, we thank the Reviewer for his/her insight comments. The title is changed into “Multi-prismatic Hollow Cube CeVO4 with Adjustable Wall Thickness Directed Towards Photocatalytic CO2 Reduction to CO”.

  1. The Abstract reflects the approach of the study, nevertheless, the main results of the work also should be presented here. The unit umol/g cat h is a little bit confusing. It is clear, that “cat” means the catalysts, but it is seems as a unit. g(cat) or other may be more understandable

Reponse: We thank the Reviewer for his/her kind comment. The main results of the study are presented as shown in the abstract. The detail contents containing wall thickness and photocatalytic activity are elaborated. The unit μmol/g cat h is changed into μmol g-1 h-1 and also is calibrated below.

  1. The section Introduction presents the important points of the topic, contains references related to the work, and reveal the importance and novelty of the work.

p 2, end of paragraph 2: “… while plastic microstructures …” Why do you mention plastics?

The mechanism of CO2 reduction should be presented in more detailed.

Reponse: We thank the Reviewer for his/her insight comments. The “plastic microstructures” is referred as structures easily artificially modulated. For understanding easily, “plastic microstructures” is changed into “artificial microstructures”.

The mechanism of CO2 reduction is presented in the section 3.2.

  1. The section Experimental.  The experimental design is appropriate; it is generally well described. However, the experimental circumstances should be presented in more detailed in all cases; moreover, the description of photocatalytic reactor also should be improved. What was the volume of the reactor, what was the pressure? What was the column, carrier gas and the detector of the GC? What was the composition of the CO2 containing gas? Was it pure CO2?

Reponse: We thank the Reviewer for his/her insight comments. The experimental circumstances are revised in page 5. The volume of reactor is added and the implemented pressure is under 1 atm. The column, carrier gas and the detector of the GC are improved. The CO2 gas is highly pure and the purity is added.

  1. In the section Results and discussion, authors describe the results shown in the corresponding figures, and tables. The figures are nice and informative.

Have you detected other products than CO?

Could you provide a more detailed reaction mechanism or equations? What is the fate of hydrogen (from water)?

Reponse: We thank the Reviewer for his/her insight comments. Other products may be generated with small trace (<5ppm) over GC detection limit. The mechanism equations are provided in the section 3.2. Hydrogen was not detected in the reaction and its byproduct is probably H2O2.

  1. The Conclusions section well summarizes the result of the work, however, practical applicability or importance also should be presented.

Reponse: We thank the Reviewer for his/her insight comments. The practical applicability or importance is added in the section.

  1. Conclusion and recommendation: This manuscript is recommended for publication after minor revision.

Reponse: We thank the Reviewer for his/her recommendation.